 SHORT REPORT

# The Axin scaffold protects the kinase GSK3β from cross-pathway inhibition

Maire Gavagan, Noel Jameson, Jesse G Zalatan*

Department of Chemistry, University of Washington, Seattle, United States

**Abstract** Multiple signaling pathways regulate the kinase GSK3β by inhibitory phosphorylation at Ser9, which then occupies the GSK3β priming pocket and blocks substrate binding. Since this mechanism should affect GSK3β activity toward all primed substrates, it is unclear why Ser9 phosphorylation does not affect other GSK3β-dependent pathways, such as Wnt signaling. We used biochemical reconstitution and cell culture assays to evaluate how Wnt-associated GSK3β is insulated from cross-activation by other signals. We found that the Wnt-specific scaffold protein Axin allosterically protects GSK3β from phosphorylation at Ser9 by upstream kinases, which prevents accumulation of pS9-GSK3β in the Axin•GSK3β complex. Scaffold proteins that protect bound proteins from alternative pathway reactions could provide a general mechanism to insulate signaling pathways from improper crosstalk.

## Editor's evaluation

This study presents a valuable and elegant kinetic analysis of the GSKbeta activity as a function of phosphorylation and Axin binding - providing insights into critical steps of Wnt pathway signaling. The results will be of big use to the broader signaling community. The work will be of broad interest to cell biologists and biochemists.

*For correspondence:
zalatan@uw.edu

Competing interest: The authors declare that no competing interests exist.

## Introduction

Glycogen Synthase Kinase 3β (GSK3β) is a potential therapeutic target for a range of diseases (**Beurel et al., 2015**; **Nusse and Clevers, 2017**), but targeting GSK3β is complicated because it has important roles in multiple signaling pathways (**Bhat et al., 2018**). Understanding how GSK3β is regulated by different signaling pathways could enable strategies to target distinct sub-populations of GSK3β.

Both Wnt and growth factor/insulin signaling pathways regulate GSK3β, but these pathways do not cross-activate (**Ding et al., 2000**; **McManus et al., 2005**; **Ng et al., 2009**). In Wnt signaling, the scaffold protein Axin binds GSK3β, its substrate β-catenin, and other proteins in a Wnt-specific complex called the destruction complex. Wnt signals inhibit GSK3β phosphorylation of β-catenin (**Hernández et al., 2012**; **Stamos et al., 2014**), causing β-catenin levels to rise and activate downstream transcription (**Nusse and Clevers, 2017**). Axin regulates kinase activity in the destruction complex, providing a mechanism to inhibit Wnt-associated GSK3β without affecting other GSK3β-dependent pathways (**Beurel et al., 2015**; **Gavagan et al., 2020**). In contrast, in growth factor/insulin signaling, the kinases PKA and PKB/Akt phosphorylate GSK3β at Ser9 (**Cross et al., 1995**; **Fang et al., 2000**; **Jensen et al., 2007**; **Sutherland et al., 1993**), which inhibits GSK3β by binding in the priming pocket and blocking substrate binding (**Dajani et al., 2001**; **Frame and Cohen, 2001**; **Stamos et al., 2014**; **ter Haar et al., 2001**; **Figure 1A**). It remains unclear why growth factor/insulin signaling does not globally inhibit GSK3β and cross-activate the Wnt pathway.

Previous work in the field suggests two potential biochemical mechanisms that could insulate Wnt signaling from insulin and growth factor signals. First, by tethering GSK3β and the Wnt substrate β-catenin together, the Axin scaffold could rescue enzyme activity from the inhibitory effects of Ser9

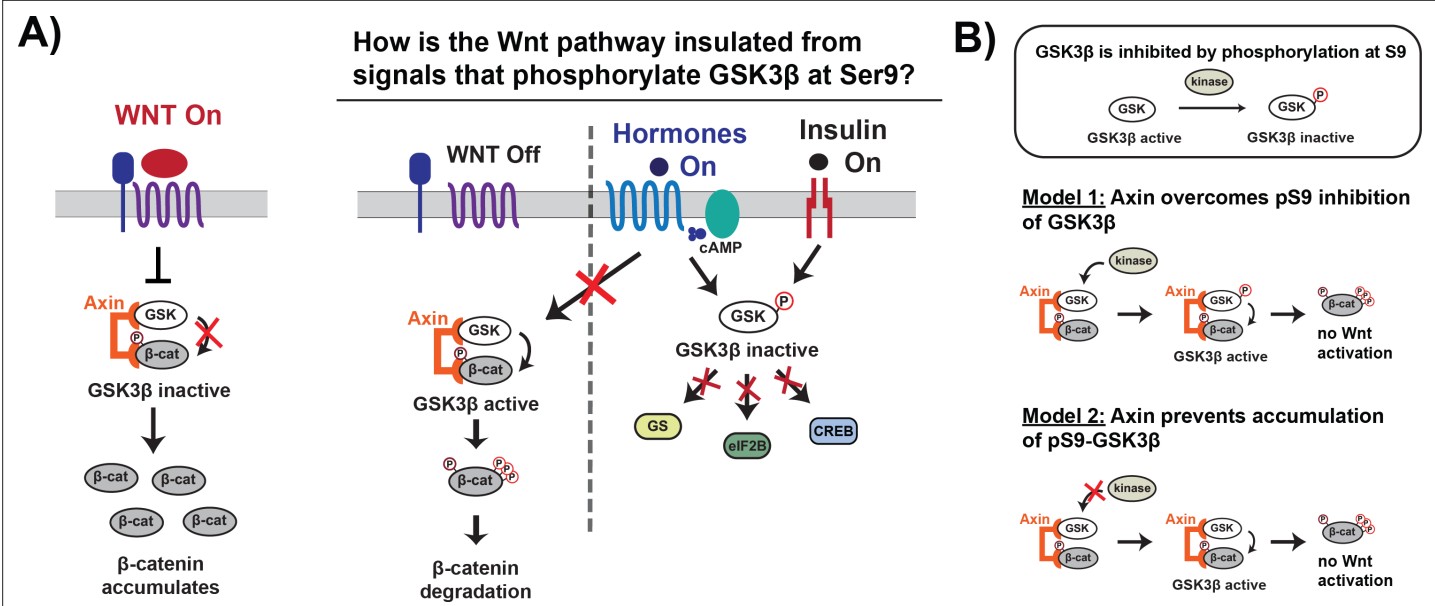

**Figure 1.** Wnt signaling is insulated from signals that phosphorylate GSK3β at Ser9. (**A**) In the Wnt pathway, the scaffold protein Axin coordinates a GSK3β complex that phosphorylates β-catenin, which is then degraded. Wnt signals inhibit phosphorylation, allowing β-catenin levels to rise and initiate a transcriptional program (*Nusse and Clevers, 2017*). In other signaling pathways, upstream signals regulate GSK3β through phosphorylation at Ser9, which blocks substrate binding, inhibits activity toward downstream substrates, and activates downstream signaling (*Sutherland, 2011*). (**B**) The scaffold protein Axin could insulate Wnt-associated GSK3β from Ser9 inhibition by restoring GSK3β activity toward β-catenin even when phosphorylated at Ser9 (Model 1) or by preventing accumulation of pS9-GSK3β in the Wnt destruction complex (Model 2).

phosphorylation (*Figure 1B*; *Beurel et al., 2015*; *Frame and Cohen, 2001*). A second possibility is that Axin prevents accumulation of pS9-GSK3β, either through direct steric effects or indirect allosteric effects (*Figure 1B*). This model is supported by in vivo experiments showing that in insulin-treated cells, Ser9 phosphorylation increases in the total GSK3β population but is unchanged in the Axin-associated GSK3β pool (*Ding et al., 2000*; *Ng et al., 2009*). Using a reconstituted biochemical system, we found that Axin allosterically protects GSK3β from phosphorylation at Ser9. The ability of scaffold proteins to allosterically regulate bound enzymes and substrates is well-established (*Good et al., 2011*), but the use of similar mechanisms to prevent competing, scaffold-independent signaling reactions has not previously been characterized. Our findings suggest a new mechanism for how scaffold proteins can promote specificity in interconnected signaling networks by shielding bound proteins.

## Results and Discussion
### Phosphorylation at Ser9 inhibits GSK3β

It is well-established that Ser9 phosphorylation inhibits GSK3β activity, but quantitative measurements are limited and variable (*Frame and Cohen, 2001*; *Stambolic and Woodgett, 1994*; *Sutherland et al., 1993*). To assess if the Wnt pathway can overcome Ser9 phosphorylation, we need quantitative metrics for comparison. We therefore used a biochemically reconstituted system to measure initial rates for the GSK3β reaction with pS45-β-catenin and determined the steady state kinetic parameters $k_{cat}$, $K_M$, and $k_{cat}/K_M$ as described previously (*Gavagan et al., 2020*). Comparing these parameters can distinguish whether Ser9 phosphorylation affects accumulation of the kinase-substrate complex or catalytic turnover. We used PKA to prepare fully-phosphorylated pS9-GSK3β (see Methods and *Figure 2—figure supplement 1C*). We observed that phosphorylation of GSK3β at Ser9 decreases $k_{cat}/K_M$ toward pS45-β-catenin by a factor of ~200-fold compared with unphosphorylated or mutant S9A GSK3β (*Figure 2*). The observed rates for unphosphorylated GSK3β and PKA-treated GSK3β_S9A are indistinguishable, indicating that the large rate decrease in pS9-GSK3β is from phosphorylation at Ser9, not any other unknown PKA phosphosites. pS9-GSK3β does not detectably saturate at high

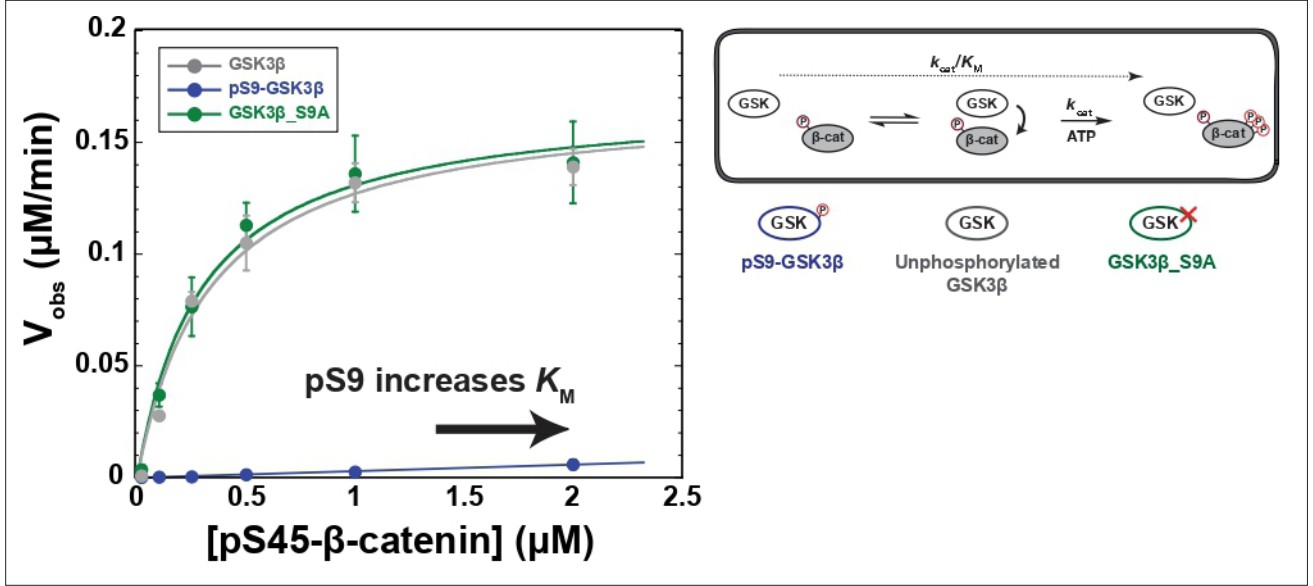

**Figure 2.** Phosphorylation at Ser9 inhibits GSK3β activity toward pS45-β-catenin. Kinetic scheme and Michaelis-Menten plots for reactions of unphosphorylated GSK3β, pS9-GSK3β, and GSK3β_S9A with pS45-β-catenin. Plots are $V_{obs}$ versus [pS45-β-catenin] at 10 nM GSK3β. GSK3β phosphorylates pS45-β-catenin at three sites: S33, S37, and T41. Values are mean ± SD for at least three biological replicates. See **Supplementary file 1a** for values of fitted kinetic parameters. See **Figure 2—figure supplement 1C and D** and **Figure 2—figure supplement 9** for characterization of GSK3β phosphorylation states and mutants.

The online version of this article includes the following source data and figure supplement(s) for figure 2:

**Source data 1.** Observed rates for data plotted in **Figure 2**.

**Figure supplement 1.** Characterization of purified proteins.

**Figure supplement 1—source data 1.** Coomassie-stained SDS-PAGE gel of purified proteins.

**Figure supplement 1—source data 2.** Western blot for phosphorylation state of GSK3β at Ser9 and Tyr216.

**Figure supplement 2.** Protein phosphorylation kinetic assays.

**Figure supplement 2—source data 1.** Representative western blots for the reaction of pS9-GSK3β with pS45-β-catenin in the presence and absence of Axin.

**Figure supplement 2—source data 2.** Representative western blots for the reaction of PKA with GSK3β in the presence and absence of Axin.

**Figure supplement 2—source data 3.** Representative western blots for the reaction of PKA with CREB$_{127-135}$ in the presence and absence of Axin.

**Figure supplement 3.** The concentration of ATP used for quantitative kinetic experiments (100 µM) is saturating for all reactions.

**Figure supplement 4.** Representative western blots for the reaction of unphosphorylated GSK3β with pS45-β-catenin in the presence and absence of Axin.

**Figure supplement 4—source data 1.** Representative western blots for the reaction of unphosphorylated GSK3β with pS45-β-catenin in the presence and absence of Axin.

**Figure supplement 5.** Plots of product vs. time for reaction of GSK3β with pS45-β-catenin in the presence and absence of Axin.

**Figure supplement 6.** Representative western blots for the reaction of pS9-GSK3β with pS45-β-catenin in the presence and absence of Axin.

**Figure supplement 6—source data 1.** Representative western blots for the reaction of pS9-GSK3β with pS45-β-catenin in the presence and absence of Axin.

**Figure supplement 7.** Plots of product vs. time for reaction of pS9-GSK3β with pS45-β-catenin in the presence and absence of Axin.

**Figure supplement 8.** $V_{obs}$ vs. [enzyme].

**Figure supplement 9.** Recombinant GSK3β is phosphorylated on multiple sites.

**Figure supplement 9—source data 1.** Recombinant GSK3β is phosphorylated on multiple sites.

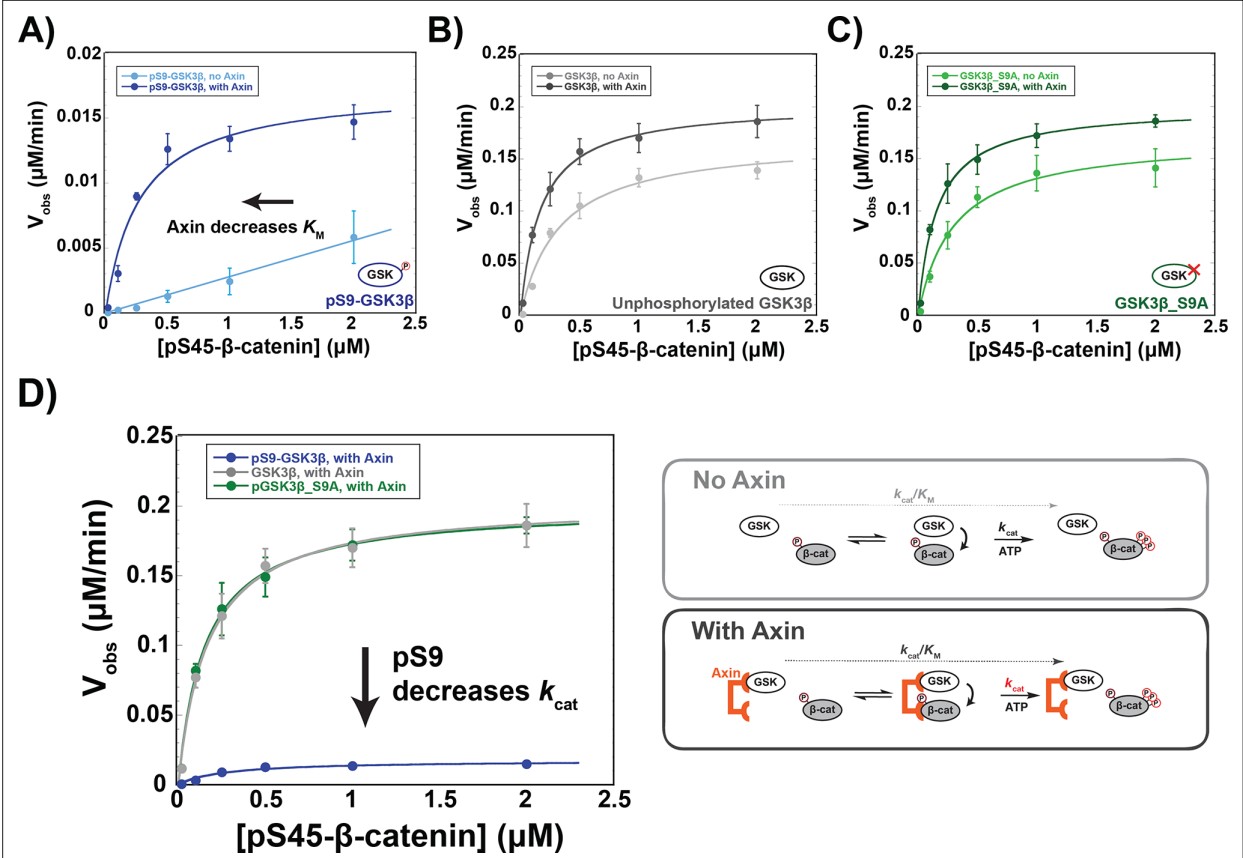

**Figure 3.** Axin restores the $K_M$ for β-catenin but cannot overcome pS9-GSK3β inactivation. (**A–C**) Michaelis-Menten plots of $V_{obs}$ versus [pS45-β-catenin] in the presence and absence of 500 nM Axin with 10 nM pS9-GSK3β (**A**), unphosphorylated GSK3β (**B**) or PKA-treated GSK3β_S9A (**C**). At the Axin concentrations used in these experiments all the GSK3β is bound to Axin (*Gavagan et al., 2020*). (**D**) Minimal kinetic scheme and Michaelis-Menten plots for reactions of GSK3β with pS45-β-catenin in the presence of Axin plotted on the same scale. Values are mean ± SD for at least three biological replicates. See *Supplementary file 1a* for values of fitted kinetic parameters.

The online version of this article includes the following source data and figure supplement(s) for figure 3:

**Source data 1.** Observed rates for data plotted in *Figure 3*.

**Figure supplement 1.** Varying the concentration of Axin does not produce larger rate effects than observed with 500 nM Axin.

substrate concentration, giving a limit for the $K_M$ of ≥2 μM. The >sevenfold increase in $K_M$ is consistent with the model that Ser9 phosphorylation inhibits GSK3β by interfering with substrate binding, although we cannot rule out the possibility that pSer9 also affects other catalytic steps.

## The scaffold protein Axin cannot overcome pS9-GSK3β inhibition

Addition of Axin to reactions with unphosphorylated GSK3β and PKA-treated GSK3β_S9A produced modest ~twofold increases in $k_{cat}/K_M$ arising from small changes to both $k_{cat}$ and $K_M$, (*Figure 3*), consistent with previous results (*Gavagan et al., 2020*). In the pS9-GSK3β reaction, however, Axin produced a ~20-fold $k_{cat}/K_M$ increase. Notably, this effect is primarily due to a decrease in the $K_M$ to 0.27 μM (*Figure 3A*), similar to the values for unphosphorylated GSK3β and GSK3β_S9A (*Figure 3B & C* and *Supplementary file 1a*). This result suggests that Axin can compensate for the inhibitory effect of pS9-GSK3β on substrate binding, possibly because the Axin binding site for β-catenin allows formation of an Axin•GSK3β•β-catenin ternary complex even when the GSK3β priming pocket is blocked.

Although Axin appears to fully rescue the $K_M$ effect from Ser9 phosphorylation, there is still a substantial ~10-fold $k_{cat}$ decrease. This behavior is consistent with a non-productive binding model (*Fersht, 1998*), in which Axin assembles a pS9-GSK3β•pS45-β-catenin complex that is still inhibited by pSer9 occupying the priming pocket. In this model, Axin-mediated assembly of the kinase-substrate complex reduces the $K_M$, but this complex cannot react so $k_{cat}$ remains impaired. In cells, if pS9-GSK3β

accumulates in the Axin-mediated destruction complex, β-catenin phosphorylation will be inhibited by ~10-fold, likely leading to improper activation of the Wnt pathway. Stimulation with high levels of Wnt ligand produces ~fivefold decreases in GSK3β phosphorylation of β-catenin (*Hannoush, 2008*; *Hernández et al., 2012*), and in vivo changes in β-catenin levels as low as twofold can have measurable effects on transcription of Wnt output genes (*Jacobsen et al., 2016*).

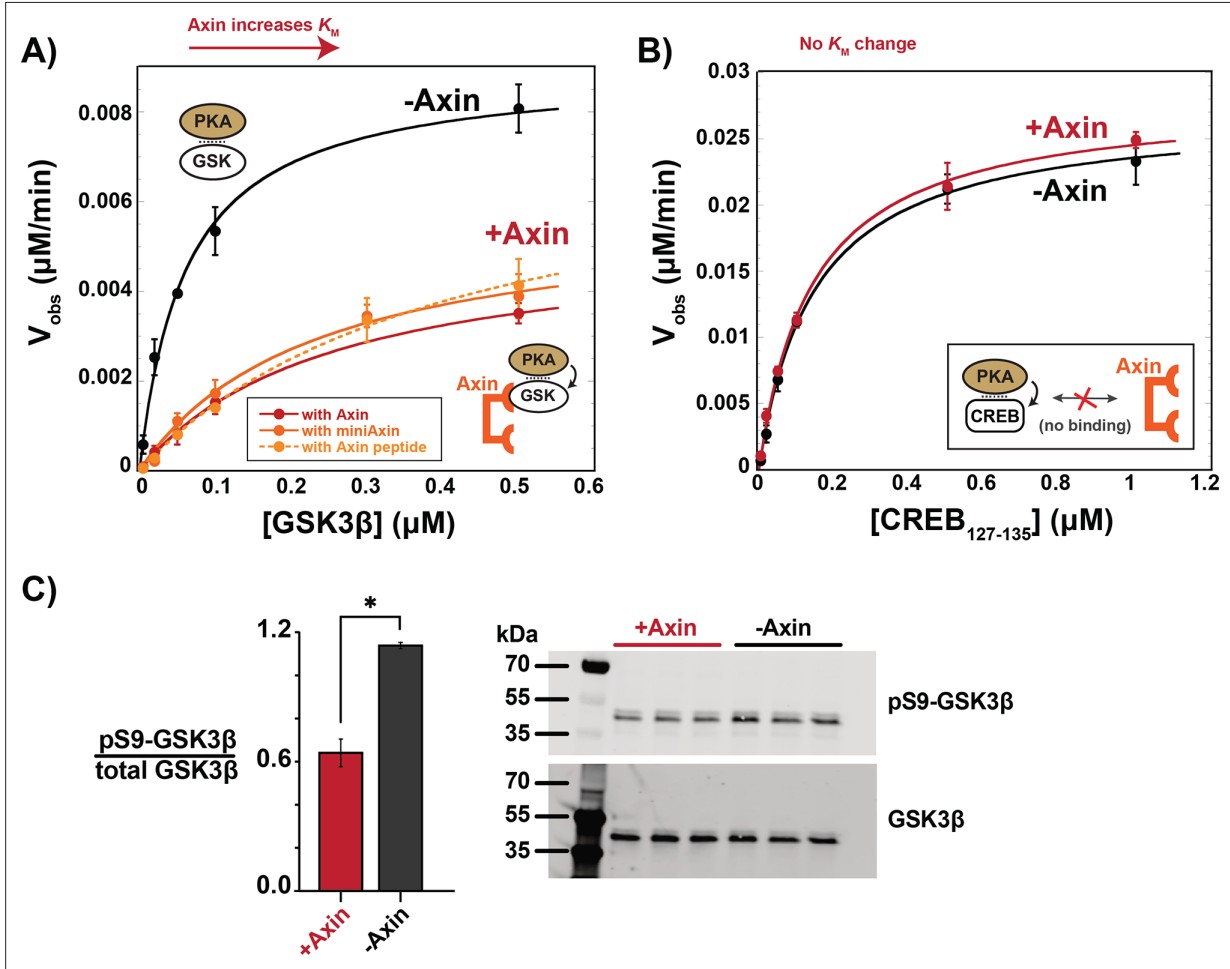

**Figure 4.** Axin prevents phosphorylation of GSK3β at Ser9. (**A**) Michaelis-Menten plots of $V_{obs}$ versus [GSK3β] with 20 nM PKA in the presence and absence of 500 nM Axin. (**B**) Michaelis-Menten plots of $V_{obs}$ versus [CREB$_{127-135}$] with 20 nM PKA in the presence and absence of 500 nM Axin. Values are mean ± SD for at least three biological replicates. See ***Supplementary file 1b*** for values of fitted kinetic parameters from (**A**) and (**B**). (**C**) Normalized western blot analysis and blot images of pS9-GSK3β in HEK293 cells transiently expressing Axin or a negative control (see Materials and methods). Normalized pS9-GSK3β levels were calculated for each biological sample by dividing pS9-GSK3β signal by total GSK3β and then averaging across three biological replicates. The p-value between Axin-expressing cells and non-Axin negative control cells is 0.00356 (two-tailed unpaired t-test). Full, uncropped western blot images are shown in ***Figure 4—figure supplement 3***.

The online version of this article includes the following source data and figure supplement(s) for figure 4:

**Source data 1.** Observed rates for data plotted in ***Figure 4A and B*** and quantification of western blot band intensities for ***Figure 4C***.

**Source data 2.** Uncropped western blots for pS9-GSK3β in HEK293 cells.

**Figure supplement 1.** Representative western blots for the reaction of PKA with GSK3β in the presence and absence of Axin.

**Figure supplement 1—source data 1.** Representative western blots for the reaction of PKA with GSK3β in the presence and absence of Axin.

**Figure supplement 2.** Plots of product vs. time for reaction of PKA with GSK3β in the presence and absence of Axin.

**Figure supplement 3.** Uncropped western blots for pS9-GSK3β in HEK293 cells.

**Figure supplement 3—source data 1.** Uncropped western blots for pS9-GSK3β in HEK293 cells.

**Figure supplement 4.** The Axin peptide is too far from the N-terminus of GSK3β to sterically occlude the Ser9 phosphorylation site.

## Axin prevents accumulation of pS9-GSK3β in the destruction complex

To test if Axin-bound GSK3β is shielded from upstream kinases, we evaluated the effect of Axin on PKA, a kinase upstream of GSK3β in growth factor signaling (*Fang et al., 2000*). We found that Axin produced a 7-fold drop in $k_{cat}/K_M$ for PKA phosphorylation of GSK3β at Ser9, primarily from a fourfold increase in $K_M$ (*Figure 4A*). This $K_M$ increase suggests that Axin interferes with formation of the PKA•GSK3β complex. To determine if this effect is specific to GSK3β, we measured the effect of Axin on the reaction with another PKA substrate, CREB (*Naqvi et al., 2014*). Axin has no effect on observed rates or $k_{cat}/K_M$ for the CREB reaction (*Figure 4B*). This result indicates that Axin is not a competitive inhibitor of PKA at the concentrations used in our assays, nor is Axin directly binding PKA to regulate its activity.

There are several plausible models for how the Axin•GSK3β interaction might disrupt PKA phosphorylation. The simplest model is that Axin sterically occludes upstream kinases from accessing the Ser9 site on GSK3β. The Axin binding site on GSK3β does not directly overlap with Ser9, nor is it immediately adjacent (*Ikeda et al., 1998*), but Axin is a large, disordered protein and could potentially extend toward the N-terminus of GSK3β. Another possibility is that Axin binding to GSK3β produces allosteric changes that make the GSK3β N-terminus less accessible to upstream kinases.

To distinguish between these possible models, we first measured the effect of a truncated Axin scaffold on PKA phosphorylation of GSK3β. This miniAxin scaffold (residues 384–518) binds GSK3β with a similar affinity as full-length Axin$_{1-826}$ (*Gavagan et al., 2020*). Addition of miniAxin produced a sixfold decrease in $k_{cat}/K_M$ with a fourfold increase in $K_M$, similar to full length Axin (*Figure 4A*). Although miniAxin is substantially smaller than full-length Axin, this construct does not allow us to conclusively rule out a steric blocking mechanism. We therefore expressed a minimal, 27 amino acid peptide, Axin$_{381-407}$, that binds GSK3β (*Hedgepeth et al., 1999*; *Stamos et al., 2014*) and cannot sterically occlude the GSK3β phosphorylation site (*Stamos et al., 2014*; *Figure 4—figure supplement 4*). Axin$_{381-407}$ produced an eightfold decrease in $k_{cat}/K_M$, with a sevenfold increase in $K_M$, similar to full length Axin and miniAxin (*Figure 4A*). This result supports the model that Axin binding leads to allosteric changes in GSK3β that make it less accessible to PKA.

One possible structural explanation for the functional data is that the Ser9 phosphorylation site adopts an inaccessible conformation when Axin binds GSK3β. However, Ser9 is located in a flexible N-terminal tail that is unresolved in x-ray crystal structures with or without Axin bound, providing no clear indication for structural changes upon Axin binding (*Dajani et al., 2003*; *Dajani et al., 2001*; *Stamos et al., 2014*; *ter Haar et al., 2001*). An alternative possibility is that the N-terminal tail remains disordered upon Axin binding but adopts altered conformational dynamics that prevent kinase access. Distinct conformational states in intrinsically disordered peptides have been observed previously (*Choi et al., 2011*; *Clouser et al., 2019*). Potentially similar functional behavior has been observed for the activation loop in the MAPK Fus3, which requires the Ste5 scaffold for phosphorylation (*Good et al., 2009*). Free Fus3 is inaccessible to its upstream kinase even though the activation loop is flexible and unresolved in x-ray crystal structures (*Reményi et al., 2005*).

To determine if Axin prevents pS9-GSK3β accumulation in vivo, we overexpressed Axin in HEK293 cells. In the absence of Axin overexpression, we observed significant basal levels of pS9-GSK3β, in agreement with previous results (*Cantoria et al., 2023*; *Fang et al., 2000*; *Whiting et al., 2015*). When Axin is overexpressed, we observed a significant decrease in pS9-GSK3β levels (*Figure 4C*), consistent with our observation that Axin prevents phosphorylation of Ser9 in vitro.

## Conclusions

The observation that Axin protects GSK3β from PKA phosphorylation in vitro is consistent with previous in vivo co-immunoprecipitation experiments suggesting that Axin-associated GSK3β is not phosphorylated at Ser9 (*Ng et al., 2009*). Beyond Axin-mediated shielding of GSK3β, other mechanisms could also contribute to preventing accumulation of pS9-GSK3β in the Wnt destruction complex. Axin interacts with the phosphatase PP2A and may promote PP2A-mediated dephosphorylation of pS9-GSK3β (*Cantoria et al., 2023*). Alternatively, subcellular localization or phase separation could sequester GSK3β in distinct pools that are associated with different signaling pathways and independently regulated (*Anton et al., 2022*; *Bock et al., 2020*; *Su et al., 2016*; *Zhang et al., 2020*). Wnt pathway proteins also phase separate (*Nong et al., 2021*; *Schaefer et al., 2018*), which could exclude kinases like Akt and PKA from accessing Wnt-associated GSK3β. Other components

of the destruction complex, such as the accessory scaffold protein APC (*Nong et al., 2021*; *Nusse and Clevers, 2017*), could contribute to these effects. Although these other mechanisms may play an important role, here we have used biochemical reconstitution to systematically evaluate two possible direct contributions of the Axin scaffold to pathway insulation, and our results suggest that Axin can allosterically control the accessibility of GSK3β to upstream signals from competing pathways. These findings provide a biochemical mechanism to explain how scaffold proteins can regulate crosstalk between interconnected cell signaling pathways.

## Materials and methods

### Protein expression constructs

The human Wnt pathway proteins GSK3β, β-catenin, and Axin (hAxin1 isoform 2, Uniprot O15169-2), along with the human CREB$_{127-135}$ peptide ILSRRPSYR and mouse PKA were cloned and expressed as previously described (*Gavagan et al., 2020*). All sequences except PKA were cloned into *E. coli* expression vectors containing an N-terminal maltose binding protein (MBP) and a C-terminal His6 tag. The catalytic subunit of mouse PKA was expressed from pET15b with an N-terminal His-tag (addgene #14921) (*Narayana et al., 1997*).

pS45-β-catenin was produced by coexpression with CK1α, as previously described (*Gavagan et al., 2020*). Lambda phosphatase ($\lambda$ PPase) was cloned with an N-terminal GST tag and a C-terminal His6 tag; the human $\lambda$ PPase sequence was obtained from VMG950 (*Good et al., 2009*). Unphosphorylated GSK3β was produced by coexpression with $\lambda$ PPase. The coexpression plasmid for GSK3β and $\lambda$ PPase was constructed by inserting the GST-$\lambda$ PPase expression cassette (without the His6 tag) into the MBP-GSK3β-His6 plasmid. GSK3β point mutants and Axin truncations were constructed by assembling PCR fragments. Unless otherwise noted, all wt and mutant GSK3β constructs in this work were coexpressed with $\lambda$ PPase to ensure they are unphosphorylated.

### Protein expression and purification

For quantitative kinetic and binding assays, all Wnt pathway, CREB, and PKA proteins were expressed in Rosetta (DE3) pLysS *E. coli* cells by inducing with 0.5 mM IPTG overnight at 18 °C. Constructs with N-terminal MBP and C-terminal His6 tags (pS45-β-catenin, Axin, and CREB$_{127-135}$) were affinity purified with HisPur Ni-NTA resin (Thermo Scientific) and amylose resin (NEB). The PKA catalytic subunit was purified on Ni-NTA resin. pS45-β-catenin was produced by coexpression with CK1α, as previously described (*Gavagan et al., 2020*). The Axin peptide, Axin$_{381-407}$ was expressed with an N-terminal MBP tag and a C-terminal His tag. The N-terminal MBP tag was removed using TEV protease in an on-bead cleavage during the purification process. Ni-NTA-purified MBP-Axin$_{381-407}$ was loaded onto amylose resin, washed with 5 column volumes of amylose wash buffer (20 mM Tris pH 8, 200 mM NaCl, 2 mM βME) and then dialyzed overnight at 4 °C in the presence of 500 nM TEV protease. The solution containing the cleaved peptide was separated from the amylose resin, concentrated, and purified by size exclusion chromatography using a Superdex 75 10/300 GL column (GE Healthcare) to remove TEV protease. Chromatography fractions were monitored by Coomassie-stained SDS-PAGE as the peptide is not detectable by UV. The final purified Axin peptide is GS-Axin$_{381-407}$-SGR-His$_6$.

Unphosphorylated GSK3β and GSK3β_S9A were produced by coexpression with lambda phosphatase and affinity purified with HisPur Ni-NTA resin (Thermo Scientific). Treatment of GSK3β and GSK3β_S9A with lambda phosphatase produces an ~fivefold increase in $k_{cat}/K_M$ that is due to an ~fivefold increase in $k_{cat}$ (*Figure 2—figure supplement 9* and *Supplementary file 1d & e*). To produce PKA-treated pS9-GSK3β and GSK3β_S9A, 10 μM of $\lambda$ PPase-treated, Ni-NTA-purified GSK3β and GSK3β_S9A were incubated with 5 μM PKA and 500 μM ATP for 2 hr at 25 °C. Phosphorylated GSK3β and GSK3β_S9A were separated from PKA by affinity purification with amylose resin (NEB). GSK3β phosphorylation at Ser9 and Tyr216 was assessed by western blot using antibodies for pSer9 GSK3β (Cell Signaling Technology #5558), pTyr216 GSK3β (BD Biosciences #612312), and MBP (Cell Signaling Technology #2396; *Figure 2—figure supplement 1*). The secondary antibodies were IRDye 800CW Goat Anti-Rabbit IgG antibody (Li-Cor #926–32211) for pSer9 and pTyr216 GSK3β and IRDye 800CW Donkey Anti-Mouse IgG antibody (Li-Cor #926–32212) for MBP. *Supplementary file 1h* includes a summary of antibodies used in this work.

Purified proteins were dialyzed into 20 mM Tris-HCl pH 8.0, 150 mM NaCl, 10% glycerol, and 2 mM DTT at 4 °C, aliquoted and stored at –80 °C. If necessary, proteins were concentrated using 10000 or 30000 MWCO Amicon Ultra-15 Centrifugal Filter devices at 4 °C, 2000×$g$. Protein concentrations were determined using a Bradford assay (Thermo Scientific). pS45-β-catenin was further purified by size exclusion chromatography using a Superdex 200 Increase 10/300 GL column (GE Healthcare) to remove a copurifying fragment before being dialyzed into 20 mM Tris-HCl pH 8.0, 150 mM NaCl, 10% glycerol, and 2 mM DTT, aliquoted, and stored at –80 °C. A Coomassie gel showing the purity of the proteins in this work is shown in *Figure 2—figure supplement 1A*.

## Quantitative kinetic assays

In vitro kinetic assays were conducted in kinase assay buffer (40 mM HEPES pH 7.4, 50 mM NaCl, 10 mM MgCl$_2$, and 0.05% IGEPAL) at 25 °C in 60 µL total volume. Reactions were initiated by adding ATP to a final concentration of 100 µM. This ATP concentration is saturating for all reactions (*Figure 2—figure supplement 3* and *Supplementary file 1c*). Reaction timepoints for initial rate kinetics were obtained at 10, 30, 60, and 90 s (pS45-β-catenin reactions with unphosphorylated GSK3β and GSK3β_S9A, *Figure 2—figure supplements 4 and 5*); 1, 2, 5, and 10 min (pS45-β-catenin reactions with pS9-GSK3β, *Figure 2—figure supplements 6 and 7*); and 0.5, 1, 2, and 4 min (vary [GSK3β] and [CREB$_{127-135}$] reactions with PKA, *Figure 4—figure supplements 1 and 2*). Ten µL aliquots were quenched by boiling in 5 X SDS loading buffer. Samples were analyzed by SDS-PAGE and quantitative western blotting as described below (*Figure 2—figure supplements 4–7* and *Figure 4—figure supplements 1 and 2*). For reactions with pS45-β-catenin, all gel samples were diluted fivefold in 1 X SDS loading buffer to prevent a gel smearing artifact that occurs with [pS45-β-catenin]≥500 nM. For reactions with PKA phosphorylation of GSK3β, samples were diluted fourfold (500 nM GSK3β reactions without Axin) or twofold (all other GSK3β concentrations) to prevent signal saturation of the western blot scan.

GSK3β-phosphorylated β-catenin was detected using a primary anti-Phospho-β-Catenin (Ser33/37/Thr41) antibody (Cell Signaling Technology #9561) that recognizes triply phosphorylated pS33/pS37/pT41-β-catenin (*Figure 2—figure supplement 2*). PKA-phosphorylated GSK3β was detected using a primary anti-phospho-GSK3β (Ser9) antibody (Cell Signaling Technology #5558) that recognizes pS9-GSK3β (*Figure 2—figure supplement 2*). PKA-phosphorylated CREB$_{127-135}$ was detected using a primary anti-phospho-CREB (Ser133) antibody (Cell Signaling Technology #9198) that recognizes pS133-CREB$_{127-135}$ (*Figure 2—figure supplement 2*). For all reactions, the secondary antibody was IRDye 800CW Goat Anti-Rabbit IgG antibody (Li-Cor #926–32211).

Concentrations of phosphorylated product in each reaction were determined by comparing western blot signal intensities to an endpoint standard containing 50 nM product phosphorylated to completion (*Figure 2—figure supplement 2*). For pS45-β-catenin reactions the endpoint is pS45-β-catenin phosphorylated to completion by GSK3β as previously described (*Gavagan et al., 2020*). The pS45-β-catenin standard was prepared in a reaction with 50 nM pS45-β-catenin, 100 nM GSK3β, and 100 µM ATP in kinase assay buffer at 25 °C for 15 min. For PKA phosphorylation of GSK3β reactions the endpoint is pS9-GSK3β, phosphorylated to completion by PKA. The pS9-GSK3β standard was prepared in a reaction with 50 nM unphosphorylated GSK3β, 100 nM PKA, and 500 µM ATP in kinase assay buffer at 25 °C for twenty-four hr. To prevent signal saturation of the western blot scan, the pS9-GSK3β standard was diluted fourfold in 1 x SDS loading dye, to a final concentration of 12.5 nM pS9-GSK3β. For CREB$_{127-135}$ reactions the endpoint is pS133-CREB$_{127-135}$, phosphorylated to completion by PKA. The pS133-CREB$_{127-135}$ standard was prepared in a reaction with 50 nM CREB$_{127-135}$, 100 nM PKA, and 200 µM ATP in kinase assay buffer at 25 °C for 20 hr.

Initial rate measurements were obtained from three independent reactions (biological replicates). Phosphorylated product levels from quantitative western blots were analyzed using Image Studio Lite 5.2.5 (Li-Cor) and kinetic parameters were determined by fitting to the Michaelis-Menten equation or to a linear equation using Kaleidagraph 4.1.3. Initial rates for each reaction were determined by fitting a linear model to a graph of [product] vs time. Kinetic parameters were determined by fitting plots of initial rates ($V_{obs}$) vs. [substrate] to the Michaelis-Menten equation $V_{obs} = k_{cat} \left[E\right]_0 \left[S\right] / \left(K_M + \left[S\right]\right)$. Standard errors for $k_{cat}$ and $K_M$ reported in *Supplementary file 1a – 1e* are from non-linear least squares fits to this equation. Standard errors for $k_{cat}/K_M$ were obtained by fitting an alternative form of the equation $V_{obs} = \left(k_{cat}/K_M\right) \left[E\right]_0 \left[S\right] / \left(1 + \left(\left[S\right]/K_M\right)\right)$. For the pS9-GSK3β reactions without Axin,

which did not detectably saturate, the value of $k_{cat}/K_M$ was obtained from the slope of linear fit to the plot of $V_{obs}$ vs. [substrate].

The reaction conditions for in vitro kinetics experiments were tested to confirm the underlying assumptions in the kinetic model. As expected, reaction rates increase linearly with increasing enzyme concentration in all reactions (*Figure 2—figure supplement 8*). We also identified the optimal scaffold concentration for all reactions (*Figure 3—figure supplement 1*). Scaffold-dependent reactions typically have optimal scaffold concentrations, and can be slow at high concentrations of scaffold protein when kinase and substrate are bound to different scaffolds (*Gavagan et al., 2020*; *Levchenko et al., 2000*; *Figure 3—figure supplement 1*).

## Phos-tag gel analysis of GSK3β phosphorylation

Phos-tag gels were prepared and run as previously described (*Gavagan et al., 2020*). After electrophoresis, the gel was incubated 3 x with transfer buffer +10 mM EDTA for 10 min before the transfer to increase transfer efficiency. Protein levels were detected with anti-MBP antibody (Cell Signaling Technology #2396; *Figure 2—figure supplement 9A*). The secondary antibody was IRDye 800CW Donkey Anti-Mouse IgG antibody (Li-Cor #926–32212).

## Cell lines

Cell culture experiments were performed with HEK293 cells (ATCC #CRL-1573, RRID:CVCL_0045). Cell cultures were tested monthly for mycoplasma contamination (Southern Biotech #13100–01).

## In vivo cell culture experiments

The Axin open reading frame (*Gavagan et al., 2020*) was cloned into the human expression vector pcDNA3.1(+) (Thermo Fisher) with a C-terminal mCherry tag. For a protein overexpression negative control, mCherry was cloned into the same pcDNA3.1(+) vector. *Supplementary file 1g* contains a summary of plasmids used in cell culture experiments.

HEK293 cells were plated at $5×10^5$ cells/mL in 10% FBS DMEM in a 24-well plate. Twenty-four hr after plating, individual wells were transfected with 50 μL Opti-MEM media containing 1 μg DNA and 1.5 μL TurboFectin (Origene TF81001). Twenty-four hr following transfection, cells were starved by replacing media with 1% FBS DMEM. Twenty-four hr after starvation, cells were treated with DMSO (2 μL in 10 mL 1% FBS DMEM). Serum starvation and DMSO treatment were included in our protocol for consistency with previously published experiments (*Cantoria et al., 2023*; *Fang et al., 2000*; *Whiting et al., 2015*). Two hr after DMSO treatment, cells were washed twice with 500 μL PBS and lysed on ice in 40 μL lysis buffer (20 mM TRIS pH 7.5, 30 mM NaCl, 20 mM NaF, 1% NP-40, 0.5% DOC, 0.1% SDS, HALT protease and phosphatase inhibitor mixture [Thermo Scientific #78440]). Lysate was then centrifuged at 13,000×$g$ for 10 min at 4 °C. The resulting supernatant was boiled for 10 min in 5 X SDS loading buffer and analyzed by SDS page and quantitative western blotting (*Figure 4—figure supplement 3*).

pS9-GSK3β was detected using the same antibody used in kinetics assays (Cell Signaling Technology #5558). Total GSK3β was detected using a primary anti-GSK3β antibody (Cell Signaling Technology #9832). The secondary antibodies were IRDye 800CW Goat Anti-Rabbit IgG antibody (Li-Cor #926–32211) for pS9-GSK3β and IRDye 680RD Donkey Anti-Mouse IgG antibody (Li-Cor #926–68072) for total GSK3. pS9-GSK3β and total GSK3β levels were analyzed using Image Studio Lite 5.2.5 (Li-Cor).

## Materials availability statement

Protein expression plasmids (*Supplementary file 1f*) generated in this work are available from Addgene or upon request.

# Acknowledgements

We thank Dustin Maly, John Scott, Maryanne Kihiu and members of the Zalatan group for comments and discussion. This work was supported by NIH R35 GM124773 (JGZ).

## Additional information

### Funding

| Funder | Grant reference number | Author |
|---|---|---|
| National Institutes of Health | R35 GM124773 | Maire Gavagan<br>Noel Jameson<br>Jesse G Zalatan |

The funders had no role in study design, data collection and interpretation, or the decision to submit the work for publication.

### Author contributions

Maire Gavagan, Noel Jameson, Conceptualization, Investigation, Visualization, Writing – original draft, Writing – review and editing; Jesse G Zalatan, Conceptualization, Supervision, Funding acquisition, Visualization, Writing – original draft, Project administration, Writing – review and editing

### Author ORCIDs

Maire Gavagan http://orcid.org/0000-0003-3986-6760
Noel Jameson http://orcid.org/0000-0001-9231-3765
Jesse G Zalatan https://orcid.org/0000-0002-1458-0654

### Decision letter and Author response

Decision letter https://doi.org/10.7554/eLife.85444.sa1
Author response https://doi.org/10.7554/eLife.85444.sa2

---

# Additional files

## Supplementary files

• Supplementary file 1. For all fitted kinetic parameters, see the Methods section for the kinetic model used to fit the data to obtain observed values of $k_{cat}$, $K_M$, and $k_{cat}/K_M$. Standard errors are from non-linear least squares fits to the initial rate data as described in the Methods. (**a**) Kinetic parameters for GSK3β reactions with pS45-β-catenin, related to *Figures 2–3*. a See *Figures 2 and 3* for data. The pS9-GSK3β reactions in the absence of Axin did not detectably saturate up to 2 μM substrate (*Figures 2 and 3*), and only the value of $k_{cat}/K_M$ could be accurately determined. No deviation from linearity was observed at 2 μM pS45-β-catenin (the highest pS45-β-catenin concentration tested), suggesting a conservative estimate that $K_M \geq 2$ μM. pS9-GSK3β, GSK3β, and GSK3β_S9A were coexpressed with lambda phosphatase. pS9-GSK3β and GSK3β_S9A were incubated with PKA and ATP before use (see Methods). The $k_{cat}/K_M$ for λ PPase-treated GSK3β is ~5 fold higher than for non-λ PPase-treated GSK3β used in previous studies (*Gavagan et al., 2020*) (see *Figure 2—figure supplement 9C* and e). (**b**) Kinetic parameters for PKA reactions, related to *Figure 4*. (**c**) $K_{M, ATP}$ values for all reactions, related to Figure S3. (**d**) Kinetic parameters for pS45-β-catenin reactions with non-PKA treated GSK3β_S9A with and without λ PPase treatment, related to *Figure 2—figure supplement 9*. a λ PPase-treated GSK3β_S9A was coexpressed with lambda phosphatase before use (see Methods). Untreated GSK3β_S9A was expressed without lambda phosphatase. (**e**) Values of $k_{cat}/K_M$ for untreated, λ PPase-treated, and PKA-treated GSK3β and GSK3β_S9A in reactions with the substrate pS45-β-catenin, related to *Figure 2—figure supplement 9*. a See (*Gavagan et al., 2020*) and *Figures 2 and 3*, and *Figure 2—figure supplement 9* for data. λ PPase-treated GSK3β and GSK3β_S9A were coexpressed with lambda phosphatase before use. PKA-treated GSK3β and GSK3β_S9A were coexpressed with lambda phosphatase and then incubated with PKA and ATP before use (see Methods). PKA-treated GSK3β is pS9-GSK3β. The $k_{cat}/K_M$ value for untreated GSK3β is from previous work (*Gavagan et al., 2020*). (**f**) Protein expression plasmids, related to Methods. a All proteins are human sequences except PKA, which is the mouse sequence. b pMBP-MG is a modified version of pMAL-p2X (New England Biolabs) with an N-terminal TEV-cleavable MBP tag and a C-terminal His₆ tag. pBH4 is a modified version of pET15b (Novagen) with an N-terminal TEV-cleavable His₆ tag. pBH4 and pMBP-MG were described previously (*Good et al., 2009*). c pMG024 was constructed by inserting GST from pETARA (*Good et al., 2009*) and the λ PPase expression cassette (without the His tag) from pMG026 into the pES001 backbone. The GSK3β_S9A mutant (pMG071) was cloned into this dual expression cassette. d pMG051 was constructed

by inserting the GST-CK1α expression cassette (without the His tag) from pMG046 into the pEF019 backbone. (g) Plasmids for cell culture experiments, related to Methods. (**h**) Antibodies, related to Methods.

- MDAR checklist
- Source data 1. Tables of data plotted in supplemental figures.

### Data availability

All data generated or analyzed during this study are included in the manuscript and supporting file; Source Data files have been provided for Figures 2-4 and supplemental figures.

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
