## [Editor Report]

This study presents a valuable and elegant kinetic analysis of the GSKbeta activity as a function of phosphorylation and Axin binding - providing insights into critical steps of Wnt pathway signaling. The results will be of big use to the broader signaling community. The work will be of broad interest to cell biologists and biochemists.

---

## [Decision Letter]

**Decision letter after peer review:**

[Editors’ note: the authors submitted for reconsideration following the decision after peer review. What follows is the decision letter after the first round of review.]

Thank you for submitting the paper "The Axin scaffold protects the kinase GSK3β from cross-pathway inhibition" for consideration by *eLife*. Your article has been reviewed by 2 peer reviewers, and the evaluation has been overseen by a Reviewing Editor and a Senior Editor. The reviewers have opted to remain anonymous.

Comments to the Authors:

We are sorry to say that, after consultation with the reviewers, we have decided that this work will not be considered for publication by *eLife* within its current scope. Both reviewers agree that the elegant kinetics analysis of GSKb activity as a function of phosphorylation and Axin binding is thorough and will be very useful for the broader scientific community. However, the main criticism shared by the reviewers, is that the study does not present a big conceptual advance in mechanistic understanding of how the Axin scaffold shields GSKb from phosphorylation and contributes to the insulation of this kinase from other activating stimuli. If you are confident that such evidence could be provided within the revisions time frame, *eLife* would welcome the resubmission and will do its best to send the revised manuscript for assessment to the same reviewers. However, if such evidence cannot be provided in a reasonable time frame, we encourage you to submit your study for consideration for publication in a more specialized journal.

*Reviewer #1 (Recommendations for the authors):*

– At the start of the Results section and in Figure 2, the definitions of kCAT and Km should be given in a bit more detail (at present the reader is only provided with a reference to the 2020 paper). This will make the story a bit easier to read for non-biochemists. For example, lines 106-109 are not intuitive to follow.

– The added value of the kinetic schemes in each of the figure panels is unclear as the link to specific biological/signal transduction steps is not made clear.

– The cartoons in Figure 1 could also be improved to clarify the questions/knowledge gaps and specific mechanistic options.

– The authors convincingly show that AXIN1 can play a role in shielding GSK3 from auto- inhibition. As it stands, the impact of this work on the field of WNT/CTNNB1 signaling is likely to remain limited. This is mainly due to the reason that the mechanism by which AXIN1 shields the WNT/CTNNB1 signaling pool of GSK3 from pSer9 inhibition remains unresolved. Based on the fact that a mini AXIN1 (i.e. an AXIN1 fragment) behaves the same as WT AXIN1, the authors conclude that AXIN1 likely causes allosteric changes on GSK3 but is less likely to block PKA from binding. They cannot conclusively show this, however, as they do not have evidence in favour of one or the other explanation.

– The impact of this work would be greater if the authors could indeed discriminate between the two possible mechanisms they present (allosteric changes on GSK3 or blocking the binding of PKA).

---

## [Author Response]

[Editors’ note: The authors appealed the original decision. What follows is the authors’ response to the first round of review.]

Reviewer #1 (Recommendations for the authors):– At the start of the Results section and in Figure 2, the definitions of kCAT and Km should be given in a bit more detail (at present the reader is only provided with a reference to the 2020 paper). This will make the story a bit easier to read for non-biochemists. For example, lines 106-109 are not intuitive to follow.

We have added additional explanation of the significance of *k*_cat_ and *K*_M_ values at the start of the Results section (lines 61-62). We have also added content and a reference to clarify lines 106-109 (now lines 83-86). Given the space restrictions for the Short Report format our explanations are brief, we request editorial guidance if further elaboration is necessary.

– The added value of the kinetic schemes in each of the figure panels is unclear as the link to specific biological/signal transduction steps is not made clear.

Kinetic schemes are included with each figure panel to enable clear comparisons between different reactions. We appreciate the reviewer’s point that it would be helpful to connect each scheme to a biological step, but we suggest that significant space would be necessary to place each reaction in a biological context and would largely reproduce information that is available in Figure 1.

– The cartoons in Figure 1 could also be improved to clarify the questions/knowledge gaps and specific mechanistic options.

We have added a panel to Figure 1 to clarify the mechanistic models and we have corrected a mistake that was noted by Reviewer #2. We would be happy to further revise the figure if the editors or reviewers have additional suggestions.

– The authors convincingly show that AXIN1 can play a role in shielding GSK3 from auto- inhibition. As it stands, the impact of this work on the field of WNT/CTNNB1 signaling is likely to remain limited. This is mainly due to the reason that the mechanism by which AXIN1 shields the WNT/CTNNB1 signaling pool of GSK3 from pSer9 inhibition remains unresolved. Based on the fact that a mini AXIN1 (i.e. an AXIN1 fragment) behaves the same as WT AXIN1, the authors conclude that AXIN1 likely causes allosteric changes on GSK3 but is less likely to block PKA from binding. They cannot conclusively show this, however, as they do not have evidence in favour of one or the other explanation.

We thank the reviewer for this important comment which details the central concern raised in the review process. To address this point, we have collected additional biochemical data that conclusively shows that the Axin shielding effect is allosteric and not a steric block. We demonstrated that a minimal, 27 amino acid Axin peptide produces the same GSK3β shielding behavior as full length Axin and miniAxin. The minimal Axin peptide does not sterically occlude the GSK3β phosphorylation site. This data is included in a revised Fig 4A and described on lines 151-157 of the revised manuscript.

– The impact of this work would be greater if the authors could indeed discriminate between the two possible mechanisms they present (allosteric changes on GSK3 or blocking the binding of PKA).

We agree with the reviewer on this central point and have conducted additional experiments that conclusively resolve the issue. These experiments are described above and included in the revised manuscript (Figure 4A, lines 115-120).